

# Using a natural capital risk register to support the funding of seagrass habitat enhancement in Plymouth Sound

Guy Hooper[1], Matthew Ashley[1], Tom Mullier[1], Martin Attrill[1], Amelia Sturgeon[2], Zoe Sydenham[3], Mark Parry[4], Katey Valentine[5] and Sian Rees[1]

[1] School of Biological and Marine Science, University of Plymouth, Plymouth, Devon, United Kingdom
[2] Tamar Estuaries Consultative Forum, Plymouth, Devon, United Kingdom
[3] Plymouth City Council, Plymouth, Devon, United Kingdom
[4] Ocean Conservation Trust, Plymouth, Devon, United Kingdom
[5] BeZero Carbon, London, United Kingdom

Corresponding author
Guy Hooper,
guy.hooper@plymouth.ac.uk

## ABSTRACT

Seagrass is an important marine habitat that provides benefits to society in the form of ecosystem services. Services include the provision of food via fisheries, the regulation of water quality and the ability to sequester and store carbon. In the UK, seagrass beds are in decline, increasing the risk of ecosystem service loss. Current efforts to protect, restore and create seagrass habitat, beyond spatial management measures, rely on grant funding and donations. Emerging carbon, biodiversity and wider ecosystem service markets offering potential revenue sources could facilitate the enhancement of seagrass habitat at scale. Participation in ecosystem service markets, requires that projects deliver on the ecosystem service benefits defined. As the benefits will have been paid for, there are risks associated with not delivering on ecosystem service benefits. It is important that the risk is clearly defined. In this study we further the marine natural capital and risk register approach and apply the method to a case study area to support the development of sustainable funding options for seagrass habitat enhancement in Plymouth Sound, UK. Habitat Suitability modelling is also used to map potential areas for seagrass habitat enhancement. We find that, in the Plymouth Sound area, the risk of loss of ecosystem services for subtidal seagrass habitats is, at present, high. This is primarily linked to the declining extent and condition of subtidal seagrass assets. Under current governance, all of Plymouth Sound's subtidal seagrass are within a Marine Protected Area, though this conservation designation does not guarantee that the seagrass bed is protected from damaging activity. Under current environmental conditions there is opportunity for widespread seagrass restoration and creation. Risk to seagrass beds and any future private funding could be reduced by governance actions that enable effective direct protection of the seagrass assets and mitigate harmful pressures (*e.g.*, reduction of water pollution). Emerging financial 'products' from seagrass ecosystem services that can support restoration and creation, require a high degree of integrity. The natural capital asset and risk register framework can provide information to underpin product development. With the development of revenue streams from ecosystem services there is a need for more intentional governance and site-based monitoring of these habitats as natural capital assets. Further research is needed to define any social or economic outcomes.
**Synthesis and Application**. By assessing the risk to the status of seagrass assets through this approach, it is possible to determine the complementary governance actions needed to underpin investment in seagrass habitat enhancement. The methods are transferable to other locations where data exists to define the asset status. These specific findings are relevant nationally where similar vectors of risk (pressures) are identified.

# INTRODUCTION

Seagrass habitats are known for generating a wide range of benefits, termed ecosystem services (ES). These include: food and biodiversity provision (*Unsworth et al., 2010*; *Bertelli & Unsworth, 2014*; *Jackson et al., 2015*; *Gamble et al., 2021*); regulation of water quality (*Moore, 2004*; *de los Santos et al., 2020*); shoreline defence (*Hansen & Reidenbach, 2012*; *Christianen et al., 2013*; *Infantes et al., 2022*) and climate regulation through carbon sequestration and storage (*Sousa et al., 2019*; *Van Katwijk et al., 2021*; *Unsworth et al., 2022*). Globally, seagrass habitat is estimated to contribute to approximately 20% of all ocean carbon sequestration despite making up only 0.1% of the sea floor (*Duarte et al., 2013*). Seagrass has subsequently been identified as an important blue carbon habitat (marine ecosystems that can store and sequester carbon) (*UK Parliament, 2021*). In the UK, the extent and condition of seagrass beds have declined by an estimated 92% over the last 100 years (*Green et al., 2021*), reducing the valuable ecosystem services delivered by these habitats for the marine environment (*Fraser & Kendrick, 2017*).

Internationally, the enhancement of seagrass habitats has been identified as a potential 'nature-based-solution', contributing to the mitigation of the future impacts of climate change and biodiversity loss (*Unsworth et al., 2022*; *Do Amaral Camara Lima et al., 2023*). Generally, seagrass habitats can be enhanced through: habitat creation (actively creating seagrass habitat in an area where it previously did not exist) (*Unsworth et al., 2019*; *Steinfurth et al., 2022*); habitat restoration (actively restoring/reintroducing seagrass habitat in an area where it once previously existed) (*Eriander et al., 2016*) and habitat recovery (passively allowing a habitat to recover where it once existed by reducing anthropogenic pressures) (*Bertelli et al., 2018*; *Branco et al., 2018*).

Currently, most projects seeking to achieve the creation, restoration or recovery of seagrass habitat rely on grant funding and philanthropic donations (*Bayraktarov et al., 2020*; *Stewart-Sinclair et al., 2020*), which, whilst an important element of this type of work, have many associated challenges. For example, this type of funding is often highly competitive, can be significantly restricted in the size and scope of activities that it can support, and operates over relatively short time horizons. Financing seagrass habitat enhancement projects that are self-sustaining and scalable is therefore challenging.

These challenges can lead to a lack of long-term support for seagrass conservation projects, limiting the amount of work that can be achieved and driving the unbalanced allocation of significant resource for fund raising to allow a project to keep moving forwards. An alternative and emerging mechanism for funding seagrass conservation involves the monetisation and sale of seagrass ES, for example as the sale of 'blue' carbon credits to the voluntary carbon market, or the sale of biodiversity credits or units (*Shilland et al., 2021*). Although still in the very early stages of development, the sale of these ES can generate a revenue source for the project and therefore a broader range of funding options can be considered. For example, repayable grants or investment with a concessional level of return may be able to be supported with this model. However, given that this type of funding relies on the ability of seagrass ES to be reliably generated and sold, investment into these types of projects inherently carries a significant degree of risk.

Participation in ecosystem service markets raises new complexities and challenges for seagrass habitat enhancement, just as it also creates new opportunities. As mentioned above there may be risk derived from the uncertainty surrounding the current state of the existing habitat and the current level of delivery for ecosystem service benefits. Without knowing the existing extent, condition and pressures faced by seagrass in an area targeted for habitat enhancement, it is more challenging to determine the appropriate technical, social and economic instruments that will lead to successful habitat enhancement. Additionally, it can bring a lack of confidence in the durability of ecosystem service delivery which may impact the project's ability to generate a revenue stream. Natural capital techniques that assess the status of habitats and their services in the context of risk may provide a useful tool in supporting decision making in this area.

The natural capital approach (NCA) relates the state of natural capital stocks (elements of nature that have value to society, such as forests, fisheries, rivers, biodiversity, land and minerals) to the flow of environmental or ecosystem services (ES) over time (*Natural Capital Committee (NCC), 2013*; *Natural Capital Committee (NCC), 2017*; *Office for National Statistics of the UK (ONS), 2017*; *Bateman & Mace, 2020*). Within a NCA, environmental features (habitats, water bodies, species populations) are considered 'natural capital assets' and are viewed as the 'stock' of resources upon which society depends (*Natural Capital Committee (NCC), 2017*). When in a healthy condition, these assets support locally, nationally, and internationally important benefits that people and societies receive from the natural environment (*Natural Capital Committee (NCC), 2017*). Termed ecosystem service benefits, these include, but are not limited to: harvested food resources, climate benefits through capture and storage of carbon, reduction of flood and storm impacts on coastal communities and maintenance of clean water and sediments. They also include availability of environments and features of interest to recreational & cultural activities and tourism (*Turner et al., 2014*).

Natural capital asset and risk registers are a decision support tool that use ecological and socio-economic data to guide management underpinning the flow of ecosystem services from natural resources (*Mace et al., 2015*; *Rees et al., 2022*). In estuarine and coastal regions, marine natural capital asset and risk registers can be used to help ensure that a wide range of benefits derived from marine natural capital assets are supported (*Rees et al., 2022*). Where

marine habitat enhancement projects are being undertaken natural capital asset and risk registers can guide decision making and resource allocation by identifying where current ecosystem service delivery is most at risk. This is achieved by determining the current state and condition of the natural capital asset and the pressures facing that asset.

Seagrass habitat assets are vulnerable and have been shown to be impacted by a number of pressures ranging from water quality to physical disturbance (*Duarte, 2002*). To credibly introduce seagrass habitat enhancement projects into ecosystem service markets the current status and risk to seagrass ecosystem service delivery must be assessed. Using a local case study, this paper aims to demonstrate how a NCA consisting of a Seagrass Asset and Risk Register can help unlock revenues from ecosystem service markets and support seagrass habitat enhancement. To achieve this, a Seagrass Asset and Risk Register was constructed for a case study area in which seagrass habitat enhancement is an objective. The asset and risk register were interpreted and discussed in the context of supporting decision making around finance options for seagrass creation, restoration and recovery in the case study area. Elementary habitat suitability mapping was used to provide further context to the site and support the prioritization of methods for habitat enhancement.

## MATERIALS AND METHODS

### Case study area

Plymouth Sound in Southwest England, United Kingdom, has a great variety of marine and estuarine habitat. In 2005, Plymouth Sound and Tamar Estuaries Special Area of Conservation (SAC), was designated to protect a number of conservation features including subtidal and intertidal seagrass species *Zostera marina* and *Z. noltei* (*Natural England, 2023*) (Fig. 1). Natural England, the statutory nature conservation body that advises the government on conservation, is obliged to report on the condition of conservation features every 6 years (*Natural England, 2020*). Through the provision of such advice, the Marine Management Organisation (MMO), Inshore Fisheries Conservation Authorities (IFCA), Local Authorities and relevant statutory Harbour Authorities manage activity and operations within and outside of MPAs, and ensure the ecological integrity of the sites (*Solandt et al., 2020*). In 2019 The Plymouth Sound National Marine Park was formally launched, with support from the UK Government's Environment Secretary. This was coupled with a declaration of intent from Plymouth City Council to work together for public benefit: "A [National] Marine Park is a specially recognised coastal or marine space important for its environment and community health and wellbeing. [National] Marine Park status will encourage greater prosperity, responsible enjoyment, deeper knowledge and enhanced appreciation of the natural world and our place within it" (*Plymouth Sound National Marine Park, 2022*).

In recent years, grants and donations have supported several conservation programmes designed to enhance subtidal seagrass habitat in the Plymouth Sound area using both active and passive interventions (*European Commission, 2022*; *Marine Conservation Society, 2022*; *ReMEDIES, 2022a*; *ReMEDIES, 2022b*). It is believed that no purposeful intervention to enhance intertidal seagrass habitat in the study area has occurred. This investigation
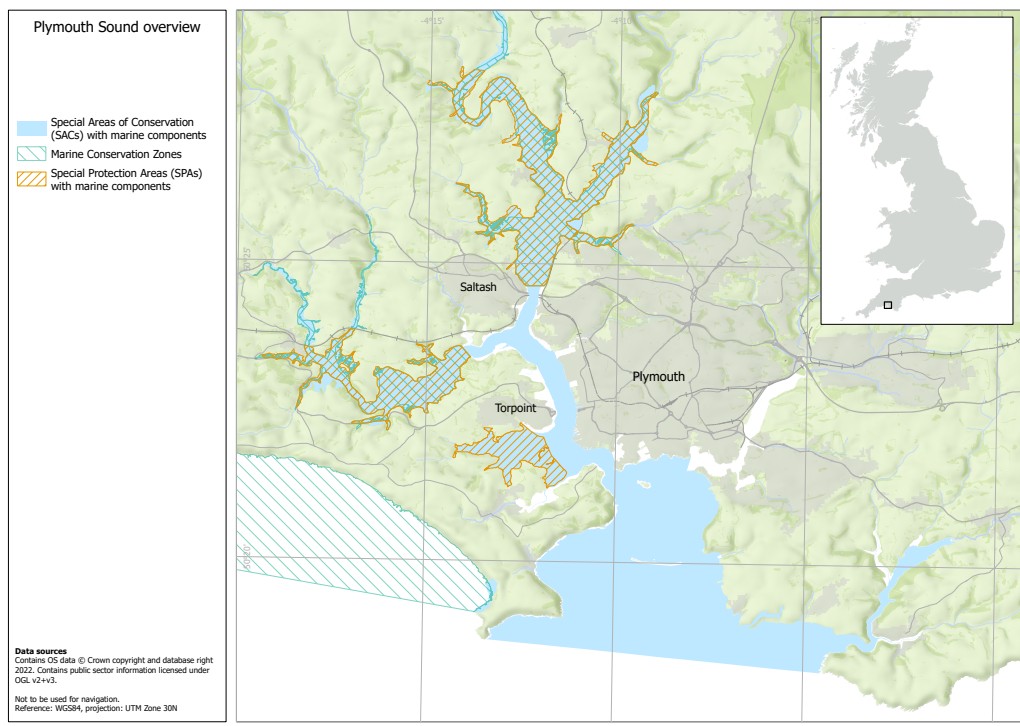

**Figure 1 Plymouth sound overview.** Map of Plymouth Sound including areas of the seabed designated for conservation as a marine protected area (Special Area of Conservation (SAC), Marine Conservation Zones (MCZ) and Special Protected Area (SPA)). Map made in ArcGIS Pro by the University of Plymouth. Data obtained from Natural England and JNCC (both Open Government License, https://naturalengland-defra.opendata.arcgis.com/maps/82bf811005484412a75c438738d51f82 and https://jncc.gov.uk/our-work/uk-marine-protected-area-datasets-for-download/).

develops a framework to understand the current state of intertidal and subtidal seagrass habitat assets in the Plymouth Sound area. The framework helps determine what actions are required to support engagement with emerging ecosystem service markets in order to maximise the benefits through ecosystem protection, restoration and creation.

## Seagrass assets

As shown in other studies, natural capital assets can be categorised into a number of biophysical assets for the marine environment, including habitat assets, species assets and the water column (*Rees et al., 2022*). In this study, habitats were the only asset class investigated. Seagrass habitat assets were identified using the European Nature Information System (EUNIS) Habitat Classification System at level 3 habitats (or above, where data existed) (*European Environment Agency, 2023*).

## The seagrass asset benefit relationship

The ecosystem services derived from seagrass habitat (*Z. marina* and *Z. noltei)* are numerous and have been defined within ecosystem service frameworks (*Finlayson, 2016*; *UK National Ecosystem Assessment. (UK NEA), 2023*) based on the habitat-ecosystem service matrix approach (*Potts et al., 2014*). From evidence identified in the literature, littoral (intertidal)

**Table 1  Contribution of seagrass habitat features to five key ES benefits.** Contribution of seagrass habitat features within the study area to five key ES benefits: wild food, sea defence, clean water and sediments, healthy climate, tourism including recreation and nature watching in the study area (ES contributions reviewed from existing studies (*Potts et al., 2014*; *Saunders et al., 2015*; *Rees et al., 2022*). The number indicates the confidence in evidence available to assign ES provision (3 = UK-related, peer-reviewed literature; 2 = Grey or overseas literature; 1 = Expert opinion).

| | Contribution to ES Good/Benefits | | | | |
| --- | --- | --- | --- | --- | --- |
| | Provisioning Service | Regulating Services | | | Cultural Services |
| Seagrass Natural Capital Assets: Seagrass Habitat in the Plymouth Sound Estuaries and Coastal Area | Food (wild food) | Clean water and sediments | Sea defence | Healthy climate | Tourism, nature watching and recreation |
| Intertidal Seagrass | 3 (Moderate) | 1 (Strong) | 1 (Moderate) | 1 (Strong) | 1 (Moderate) |
| Subtidal Seagrass | 3 (Strong) | 2 (Moderate) | 1 (Moderate) | 2 (Strong) | 1 (Moderate) |

and sublittoral (subtidal) seagrass habitats were found to provide moderate or significant contribution to all 5 key ES benefits (Table 1). Sublittoral seagrass provides a significant contribution to wild food benefits, as nursery habitats for fish and shellfish species, and significant contribution to capture and storage of carbon. Methods used to determine the Seagrass asset benefit relationship are described further in Appendix S1.

## Seagrass habitat asset register

Contribution to ES benefits from seagrass natural capital assets assumes those habitats are in healthy condition, providing the structure, function, and ecological processes to support flow of intermediate services, ecosystem services and finally the key ES goods/benefits assessed. To assess the status of seagrass natural capital assets within the Plymouth Sound area, an asset register was compiled following the methods in *Mace et al. (2015)* and *Rees et al. (2022)*. The asset register identifies the current status of seagrass (1) habitat extent, (2) condition, and (3) spatial configuration. The asset register established which assets were unlikely to be providing expected contribution to key ES benefits. Importantly the asset register assesses the ecosystem delivery for Plymouth Sound seagrass at two different levels: (1) the feature level (whole feature/total seagrass within the defined area); and where data exists (2) the individual bed level (individual seagrass beds).

## Seagrass habitat extent
### Seagrass habitat extent (feature level)

A composite habitat layer of intertidal and subtidal seagrass assets was generated from survey data provided by the European Marine Observation and Data Network (EMODnet), UKSeaMap modelled data (*Joint Nature Conservation Committee, 2018*; *EMODnet/EUSeaMap, 2019*) and by colleagues from the University of Plymouth and the Ocean Conservation Trust (see Appendix S1). The extent of intertidal and subtidal seagrass within the study area were calculated from the composite mapping data using ESRI ArcGIS Pro. The extent of overlap between conservation features and designated conservation sites was also calculated (see Appendix S1). The change in intertidal and subtidal seagrass extent was determined by looking at the relevant Natural England conservation seagrass condition assessments (*Curtis, 2012*; *Bunker & Green, 2019*).

### Seagrass habitat extent (individual bed level)

Using the same methods described above, the extent of individual seagrass beds within the study area was determined. There is currently no information on the change in extent of individual intertidal seagrass beds within the study area, highlighting this as an area for future research and data collection.

## Seagrass habitat condition

### Seagrass habitat condition (feature level)

For habitat that is within a designated conservation site, overall condition (for the SAC), can be inferred though the condition status and conservation objectives described in conservation advice packages provided by Natural England (*Natural England, 2023*). As both intertidal and subtidal seagrass are designated conservation features within the Plymouth Sound and Estuaries SAC, this approach was used to assess condition for both habitats at the 'feature level'. Likely relative condition (LRC) modelling was also used to infer condition (method described below and in Appendix S1).

*Likely relative condition modelling.* LRC is a proxy technique that uses habitat sensitivity data and data on potentially harmful pressures in the area to assess potential habitat condition. Seagrass sensitivity data from the Marine Evidence based Sensitivity Assessment, MarESA, data product (*d'Avack et al., 2022a*; *d'Avack et al., 2022b*) was used in this case. Abrasion and Penetration MarESA pressures were used for this modelling, comprising of anchoring, mooring and fishing activity (*Des Clers et al., 2008*; *Enever et al., 2017*; *Langmead et al., 2017*). In this investigation, we define our scale of LRC as 1 indicating poor LRC (the habitat has been exposed to a pressure to which it is sensitive) and 5 indicating potentially good condition, within the shifted baseline of human activity (no exposure to pressure or pressure thresholds are within the tolerances of the defined sensitivity of the habitat) (*Rees et al., 2022*). Using this technique, we were able to indirectly infer the condition of seagrass habitats at a feature level.

### Seagrass habitat condition (individual bed level)

To assess the condition of individual seagrass beds, data related to each bed was required. The latest Natural England condition assessments for subtidal seagrass measures: percentage cover; shoot density; plant length; and percentage of infected leaves at individual subtidal seagrass beds (*Curtis, 2012*; *Bunker & Green, 2019*). Where possible this data was used to assess the condition of individual subtidal seagrass beds. For intertidal seagrass there are currently no direct condition indicator measurements. Direct measurements of intertidal seagrass condition as done previously for subtidal seagrass (*Bunker & Green, 2019*) would close this data gap.

*Water quality information.* Water quality and clarity contribute to the health of seagrass. Water body status data, in reference to Water Framework Directive (WFD) targets, provides information on local water quality. The most recent water body status data (2013–2019) was accessed from His Majesty's (HM) Government online resources (*Environment Agency, 2021*) to investigate the current status and trend of the four water bodies within Plymouth

Sound that overlap with individual seagrass habitat assets. By assessing the status of water bodies in Plymouth Sound against WFD targets we may be able to partly infer the condition of individual beds within the sound. Further information on how water quality data was used to assess the condition of seagrass beds is described in Appendix S1.

## Seagrass habitat spatial configuration
### Seagrass habitat spatial configuration (feature level)
Akin to the methods used by *Ashley, Rees & Mullier (2018)*, where Conservation Advice Packages for features of conservation interest included an assessment of spatial distribution, for example, location, this was included in the risk assessment (see Appendix S1).

### Seagrass habitat spatial configuration (individual bed level)
Due to the lack of data for individual seagrass beds it was not possible to reliably infer the spatial configuration of individual seagrass beds.

## Policy targets
*Mace et al. (2015)* categorised risk according to the performance of the asset-benefit relationship to relevant policy targets. Policy targets in this context are considered to be societal aspirations for the asset-benefit relationship and, as such, forming a threshold target against which risk can be defined. For this risk register, policy targets from international and UK policy frameworks were selected for their relevance to seagrass and other benthic protected habitats (extent, condition and spatial configuration) and are outlined in Appendix S2. Where possible policy targets were applied to individual seagrass beds. Due to the context in which some policy is applied *e.g.*, water quality at a full estuary scale, it was not always possible to assess seagrass at the individual bed level.

## Risk assessment
The risk register, adapted from *Mace et al. (2015)*, introduces assessment of the risk to the asset—ES benefit relationship. Essentially, the risk register considers the nature and severity of risk to the asset—benefit relationship, in relation to the state of the asset's extent and condition. Following the process defined by *Mace et al. (2015)*, and adapted for marine systems by *Rees et al. (2022)*, the seagrass asset-benefit relationships were assessed against the complied evidence according to the identified policy targets (Appendix S2). Each component characteristic (extent, condition and spatial configuration) was assessed for status and trend (Appendix S2). Within the red, amber, green risk ratings proposed by *Mace et al. (2015)*, we applied an additional precautionary approach to identify risk as demonstrated by *Rees et al. (2022)*. In instances where the status of benefit is below target, and the trend negative, we apply an adapted amber risk rating with an asterisk to highlight those asset-benefit relationships that are at risk of tipping over to a red risk rating (*Rees et al., 2022*). Within the risk register a confidence score based on robustness and agreement of evidence (*IPCC, 2014*) linked to the trend and status, enabled confidence in results to be presented (Appendix S2). In the final asset and risk register output, lighter shaded red, amber or green cells indicates a risk rating where there is less confidence (greater uncertainty) in the rating, due to limited evidence and/or limited agreement between evidence sources (*Rees et al., 2022*) (Appendix S2).

## Mapping habitat suitability

Habitat suitability mapping was carried out to determine the maximum potential area for seagrass habitat enhancement within Plymouth Sound, additional to existing seagrass habitat assets. This habitat suitability mapping approach requires habitat preference data and environmental predictor data. Intertidal and subtidal seagrass habitat preference data, obtained from the Marine Biological Association's MarLIN habitat and species databases (*d'Avack et al., 2022a*; *d'Avack et al., 2022b*) (Appendix S1), were used to define the range of environmental conditions conducive to seagrass growth. Environmental predictor data, obtained from the sources shown in Appendix S1, are spatial data describing environmental variables, that can be assessed alongside preference data to determine habitat suitability. Underlying broad scale habitat (substrate type), bathymetry (depth), and kinetic energy at the seabed due to waves (wave exposure), were the environmental predictor variables used in this model. These variables were chosen based on data availability, their common use in seagrass habitat suitability models, and their suggested influence on seagrass presence and distribution (*Greve & Binzer, 2004*; *Bertelli et al., 2022*).

Using ESRI ArcGIS Pro, the data layers were intersected and filtered using the determined thresholds to create two habitat suitability layers. These represent the extent of habitat that could be suitable for intertidal and subtidal seagrass habitat enhancement. Existing seagrass habitats were omitted from the model to highlight where new seagrass could be recovered, restored, or created. The use of a relatively low number of environmental predictor variables kept habitat suitability outputs broad, supporting the main objective of the exercise. The benefits and limitations of this approach are discussed further in 'Mapping habitat suitability'.

# RESULTS

## Seagrass habitat extent

The current extent of seagrass habitat assets within the Plymouth Sound area, extracted from European Marine Observation and Data Network (EMODnet) composite mapping data, is 1.177 km$^2$ (0.632 km$^2$ intertidal and 0.545 km$^2$ subtidal (Fig. 2). Under current governance, 0.916 km$^2$ of seagrass habitat is within an MPA.

## Seagrass risk register

Figure 3 describes the seagrass habitat asset-benefit relationships assessed in the risk register. The intertidal and subtidal seagrass assets are columns, and the ecosystem service benefits in rows. For each ES, where possible, the seagrass asset's extent (Ex), condition (Con) and spatial configuration (Sp) is risk assessed through analysis of indicator data in relation to policy targets. This risk register was built akin to the marine risk register designed by *Rees et al. (2022)*. The colour of the cell shows the risk rating based on the scoring matrix (Appendix S2). Red indicates it is at high risk, amber at medium risk (*amber cells with an asterisk, indicate asset status is below target and the trend in status is declining, suggesting risk rating is close to moving to the high-risk category), green risk ratings are at low risk (*Rees et al., 2022*). Lighter shaded, red, amber or green cells indicates RAG risk rating where there is less confidence (greater uncertainty) in the risk rating, due to limited evidence

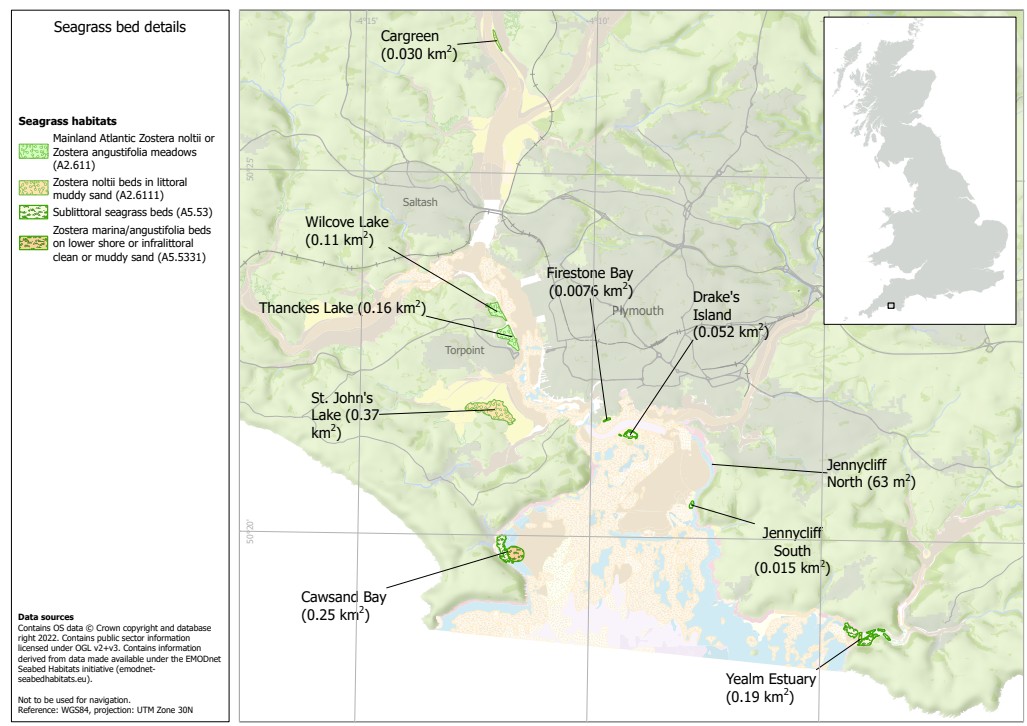

**Figure 2** **Plymouth Sound seagrass habitat assets.** Habitat extents of intertidal and subtidal seagrass assets within Plymouth Sound. Map made in ArcGIS Pro by the University of Plymouth. Habitat data obtained from UKSeaMap: JNCC (Open Government License, https://hub.jncc.gov.uk/assets/202874e5-0446-4ba7-8323-24462077561e), EMODnet Seabed Habitats Initiative (https://emodnet.ec.europa.eu/en/seabed-habitats), the University of Plymouth and the Ocean Conservation Trust.

and/or limited agreement between evidence sources (*Rees et al., 2022*). In each cell, the letter, A/B/B\*/C refers to the overall RAG rating based on status and trend. The number refers to the total uncertainty of the RAG rating based on the amount and agreement of evidence.

The risk register demonstrates that the ecosystem services from seagrass beds are at risk of loss. In total, 120 seagrass habitat asset—benefit relationships were investigated. 95 of the seagrass habitat asset—benefit relationships were given a high risk rating and 25 were given a medium risk rating. The high-risk rating seen across the majority of subtidal seagrass assets in the study area is primarily due to their unfavourable condition status in the most recent Natural England condition assessments, low LRC scores and low-grade water body statuses. The low-medium risk rating given for intertidal seagrass is largely due to the lack of evidence for intertidal seagrass status and an unknown status generated in the latest Natural England condition assessments. The full asset and risk register rationale behind the RAG rating can be found in Appendix S2. In the risk register subtidal seagrass assets are assessed at the feature and individual bed level. However, there was not enough evidence on the status of individual intertidal seagrass assets to determine a risk rating for ecosystem service delivery for each intertidal seagrass bed.

| | Feature Level | | | | | | Individual Bed Level (where data exists) | | | | | | | | | | | | | | | | | |
| | Intertidal Seagrass | | | Subtidal Seagrass | | | Firestone Bay | | | Drake's Island | | | Jennycliff North | | | Jennycliff South | | | Cawsand | | | Yealm | | |
| Ecosystem Services | Ex | Con | Sp | Ex | Con | Sp | Ex | Con | Sp | Ex | Con | Sp | Ex | Con | Sp | Ex | Con | Sp | Ex | Con | Sp | Ex | Con | Sp |
|---|---|---|---|---|---|---|---|---|---|---|---|---|---|---|---|---|---|---|---|---|---|---|---|---|
| Food (Wild food - fish and shellfish). | B(8) | B(8) | B(8) | C(4) | C(4) | C(4) | C(4) | C(4) | C(4) | C(4) | C(4) | C(4) | C(4) | C(4) | C(4) | C(4) | C(4) | C(4) | B(4) | C(4) | B(4) | C(4) | C(4) | C(4) |
| Healthy climate (carbon sequestration). | B(8) | B(8) | B(8) | C(4) | C(4) | C(4) | C(4) | C(4) | C(4) | C(4) | C(4) | C(4) | C(4) | C(4) | C(4) | C(4) | C(4) | C(4) | B(4) | C(4) | B(4) | C(4) | C(4) | C(4) |
| Sea defence (natural hazard regulation/flood prevention) | B(8) | B(8) | B(8) | C(4) | C(4) | C(4) | C(4) | C(4) | C(4) | C(4) | C(4) | C(4) | C(4) | C(4) | C(4) | C(4) | C(4) | C(4) | B(4) | C(4) | B(4) | C(4) | C(4) | C(4) |
| Tourism/nature watching | B(8) | B(8) | B(8) | C(4) | C(4) | C(4) | C(4) | C(4) | C(4) | C(4) | C(4) | C(4) | C(4) | C(4) | C(4) | C(4) | C(4) | C(4) | B(4) | C(4) | B(4) | C(4) | C(4) | C(4) |
| Clean water and sediments | B(8) | B(8) | B(8) | C(4) | C(4) | C(4) | C(4) | C(4) | C(4) | C(4) | C(4) | C(4) | C(4) | C(4) | C(4) | C(4) | C(4) | C(4) | B(4) | C(4) | B(4) | C(4) | C(4) | C(4) |

**Figure 3 Seagrass risk register.** The assets are columns and the benefits are rows. For each ecosystem service (ES) the risk was assessed through analysis of evidence in relation to policy targets. The colour of the cell shows the risk rating for the asset status extent (Ex), condition (Con) and spatial configuration (Sp). Red indicates it is at high risk, amber at medium risk (*amber cells with an asterisk, indicate asset status is below target and the trend in status is declining, suggesting risk rating is close to moving to the high risk category), green risk ratings are at low risk. Lighter shaded, red, amber or green cells indicates where there is less confidence (greater uncertainty) in the risk rating, due to limited evidence and/or limited agreement between evidence sources.

The risk of loss of ecosystem services from intertidal seagrass habitats is currently moderate. This is largely due to the 'unfavourable unknown' conservation status reported in Natural England's latest conservation objectives (*Natural England, 2023*). The 'unfavourable unknown' assessment was based on sediment contaminant levels and an ephemeral macroalgae issue (ephemeral macroalgae overlying seagrass beds, preventing primary production) (*Natural England, 2023*) in the Plymouth Sound and Estuaries Area, both of which are proxy measures for approximating the condition of intertidal seagrass. Despite this assessment, Natural England believe that the extent of intertidal seagrass within the whole site has been maintained (*Natural England, 2023*). This finding is reflected in the medium—low risk rating given to intertidal seagrass for 'extent' in the risk register. Generally, it is not clear how 'at risk' intertidal seagrass beds in the Plymouth Sound and Estuaries area are. Natural England's assessment of intertidal seagrass is graded as 'low confidence'. For this reason, a precautionary assessment of 'moderate' was chosen for the risk of loss of ecosystem services from intertidal seagrass.

The risk of loss of ecosystem services from subtidal seagrass habitats is, at present, high. Natural England assessed the conservation status of subtidal seagrass in the Plymouth Sound and Estuaries Area to be 'Unfavourable Declining' (*Natural England, 2023*). The percentage area of subtidal seagrass in 'unfavourable declining' condition across the Plymouth Sound and Estuaries site is reported to be 47.9% (*Natural England, 2023*). This was primarily due to the reduced extent, distribution and condition of multiple subtidal seagrass beds. Natural England reports the pressures most likely to be impacting the seagrass as: abrasion to the seabed, nutrient enrichment, and water clarity (*Natural England, 2023*). Likely Relative Condition modelling indicated that some subtidal seagrass assets were likely to be impacted by physical abrasion from recreational anchoring and mooring. Despite Natural England's 'unfavourable declining' rating for subtidal seagrass in the Plymouth Sound and Estuaries SAC, there have been increases in the extent of seagrass beds in Cawsand Bay (*Bunker & Green, 2019*). These measurements are reflected in the risk-ratings for extent at Cawsand seagrass beds.

## Seagrass habitat potential

Not including areas of existing intertidal seagrass, habitat suitability mapping based on the underlying habitat type, bathymetry and exposure determined that the potential area suitable for intertidal seagrass habitat enhancement was 14.3 km$^2$ (Fig. 4). Not including areas of existing subtidal seagrass, habitat suitability mapping based on the same factors, determined that the potential area available for subtidal seagrass habitat enhancement was 7.2 km$^2$ (Fig. 5).

# DISCUSSION

## Summary

Under current governance the majority of Plymouth Sound seagrass habitat assets (78%) are within a Marine Protected Area (MPA), though this nature conservation designation does not guarantee that the seagrass bed is protected from damaging activity. Furthermore, the risk register demonstrates that the risk of loss to ecosystem service delivery for all subtidal seagrass assets in the Plymouth Sound area is currently high due to pressures exerted on the habitats. These risks are primarily from water quality (historic and current) and recreational anchoring and mooring. Crucially, it highlights where further research and monitoring is required before ecosystem service markets linked to seagrass habitat enhancement are developed further. There are opportunities to protect existing seagrass habitat through more effective governance measures (*e.g.*, no anchor zones or anchoring alternatives) across all seagrass beds inside and outside current MPAs. This could passively facilitate seagrass habitat enhancement, improving seagrass extent and condition and increasing confidence in ecosystem service delivery. Habitat suitability models highlight the potential for intertidal and subtidal seagrass habitat enhancement in the area. In order to support the financial mechanisms used for future seagrass habitat enhancement, consideration must be given to the following points of discussion.

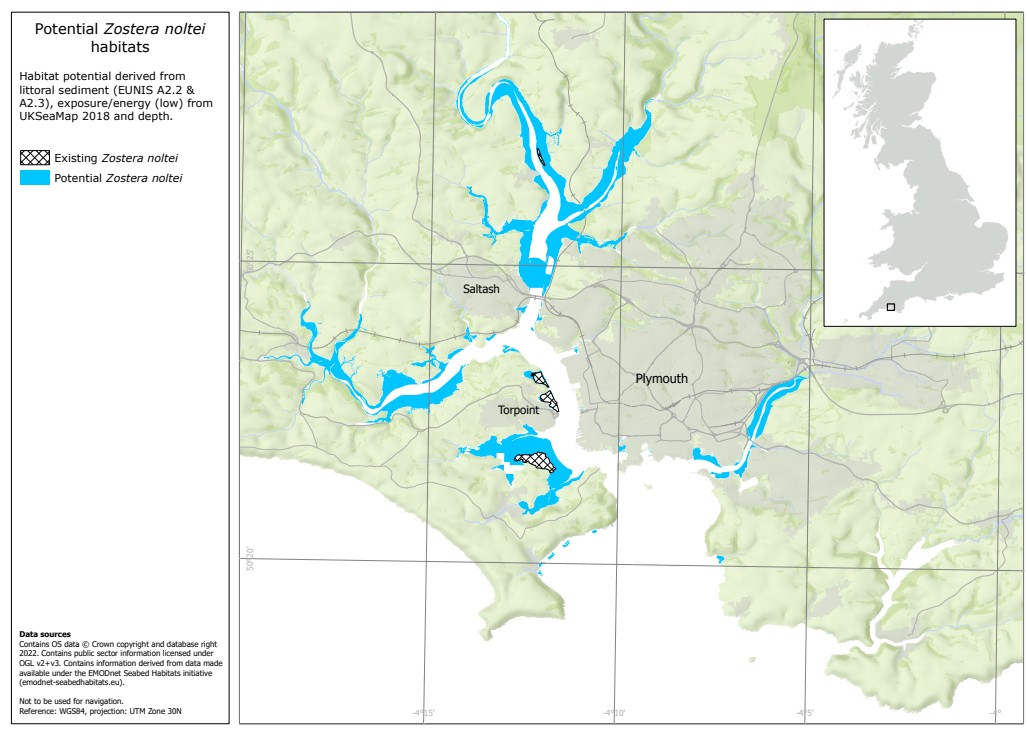

**Figure 4** **Plymouth Sound intertidal seagrass habitat potential across all intertidal sediments within the environmental variable thresholds described in Appendix S1.** Existing intertidal seagrass beds also shown. Map made in ArcGIS Pro by the University of Plymouth. Habitat data obtained from UKSeaMap: JNCC (Open Government License, https://hub.jncc.gov.uk/assets/202874e5-0446-4ba7-8323-24462077561e), EMODnet Seabed Habitats Initiative (https://emodnet.ec.europa.eu/en/seabed-habitats), the University of Plymouth and the Ocean Conservation Trust. The habitat potential data layer was created by intersecting habitat preference and environmental predictor data. Preference data was obtained from the MarLIN database (https://www.marlin.ac.uk/). Predictor data was obtained from UK SeaMap (https://hub.jncc.gov.uk/assets/202874e5-0446-4ba7-8323-24462077561e), EMODnet Seabed Habitats Initiative (https://emodnet.ec.europa.eu/en/seabed-habitats) and EDINA Digimap (https://digimap.edina.ac.uk/).

## Suitability of current monitoring data to assess natural capital asset status

The accuracy of risk ratings for ecosystem service loss generated by the asset and risk register is only as accurate as the data that is entered into the register. To reliably assess the risk of loss of ecosystem services for seagrass habitat assets, accurate and up-to-date data is required. For this asset and risk register a large proportion of the trend and condition data used was provided by Natural England, the statutory conservation body that is responsible for assessing the condition of designated conservation features. Currently, Natural England are obliged to assess the condition of designated conservation features every 6 years (*Joint Nature Conservation Committee, Natural England, 2012*; *Natural England, 2020*), though timescales have varied. Intertidal and subtidal seagrass habitat extent and condition can change rapidly over relatively short timescales (*Bertelli et al., 2018*; *Kletou et al., 2018*; *Danovaro et al., 2020*). In the Wadden Sea, aerial surveys revealed a three-to-four-fold

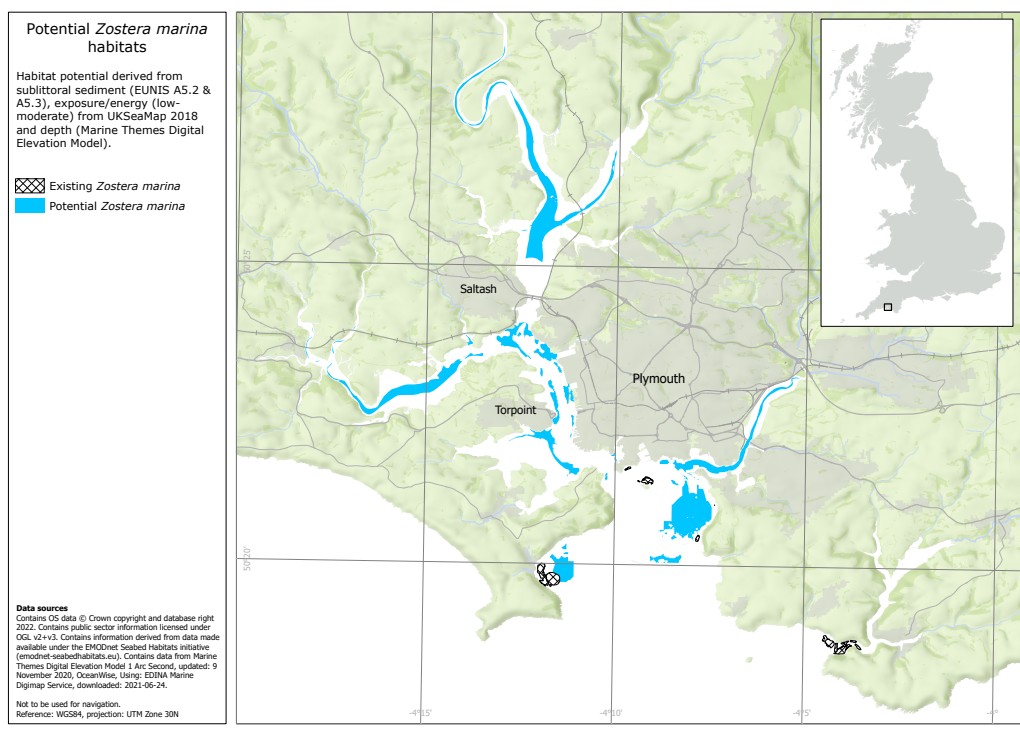

**Figure 5** **Plymouth Sound subtidal seagrass habitat potential across all subtidal sediments within the environmental variable thresholds described in Appendix S1.** Existing subtidal seagrass beds also shown. Map made in ArcGIS Pro by the University of Plymouth. Habitat data obtained from UKSeaMap: JNCC (Open Government License, https://hub.jncc.gov.uk/assets/202874e5-0446-4ba7-8323-24462077561e), EMODnet Seabed Habitats Initiative (https://emodnet.ec.europa.eu/en/seabed-habitats), the University of Plymouth and the Ocean Conservation Trust. The habitat potential data layer was created by intersecting habitat preference and environmental predictor data. Preference data was obtained from the MarLIN database (https://www.marlin.ac.uk/). Predictor data was obtained from UK SeaMap (https://hub.jncc.gov.uk/assets/202874e5-0446-4ba7-8323-24462077561e), EMODnet Seabed Habitats Initiative (https://emodnet.ec.europa.eu/en/seabed-habitats) and EDINA Digimap (https://digimap.edina.ac.uk/).

increase in the extent of *Zostera* spp., in the space of just 12 years (*Reise & Kohlus, 2008*). For seagrass habitat, 6-year monitoring cycles are not frequent enough to accurately determine the change in extent and condition over time, or to assess the response to conditions driven by the natural environment and by anthropogenic activities.

Furthermore, with a monitoring frequency of 6 years, discrepancies in techniques used to assess the extent and condition of seagrass habitat can create difficulties for data comparison and trend analysis. In the 2012 condition assessment by *Curtis (2012)*, new methods were used to measure the extent and condition of subtidal seagrass in Plymouth. Essentially the 2012 survey acts as new baseline for subtidal seagrass extent and condition monitoring in Plymouth Sound (*Bunker & Green, 2019*). In 2018, *Bunker & Green (2019)* carried out the next and most recent condition assessment. Though the methods used were the same, differences in divers used, areas surveyed, equipment, and visibility conditions have impacted the opportunity to compare results between years.

When considering opportunities for sustainable finance, risk could be reduced through seagrass monitoring that is tailored to the investment. This will require monitoring of the status of the natural capital asset (extent, condition, spatial configuration), at individual seagrass beds, that is more frequent and robust enough to detect natural and human induced change. For this to be achieved, accurate data on the dominant drivers that exert pressure on seagrass habitat, such as recreational boating and water quality, must also be integrated into the monitoring strategy. In some instances, this may require the use of deployable *in-situ* sensors that can provide data on pressures for individual seagrass beds. Emerging methods to monitor seagrass that use machine learning, remote sensing and acoustic techniques to provide rapid, cost effective and non-intrusive solutions to seagrass surveys will be key to ensuring sustainable and regular data acquisition (*Duffy et al., 2018*; *Sengupta, Ersbøll & Stockmarr, 2020*; *Ballard et al., 2021*; *Ventura et al., 2022*). Overall, from an investment perspective, risk can be reduced though effective monitoring that does not necessarily have to be cost prohibitive.

## Mapping habitat suitability

Mapping seagrass habitat suitability is useful for prioritising areas for seagrass habitat enhancement (*Hu et al., 2021*; *Bertelli et al., 2022*; *Ward et al., 2023*). In this study, an elementary habitat suitability model demonstrated 14.3 and 7.2 km$^2$ of potentially suitable habitat within Plymouth Sound for intertidal and subtidal seagrass respectively. Areas of potentially suitable habitat were identified by combining habitat preference data with three environmental predictor variable datasets (Underlying substrate, depth and wave exposure). Compared to other habitat suitability models, fewer predictor variables were used in this model (*Bertelli et al., 2022*). However, by limiting the number of predictor variables, habitat suitability outputs were broad, showing the likely maximum potential area for seagrass habitat enhancement, particularly useful in the early stages of site and method prioritisation. Replicating this method in other areas should be relatively straightforward as underlying substrate, depth and exposure are commonly available datasets (*Ban, 2009*), though this will depend on the location and scale of the site being investigated.

The main limitation when constraining the number of predictor variables is the risk of including 'unsuitable' areas within the 'suitable' habitat layers. As seagrass habitat enhancement projects develop, habitat suitability maps that are more accurate and spatially explicit may be required. One way this could be achieved is by developing a more complex model with a larger number of environmental predictor variables that eliminates a larger proportion of 'unsuitable' habitat. There are many environmental variables that influence the presence and distribution of seagrass (*Greve & Binzer, 2004*) and in theory these could all be used in a habitat suitability model. Similarly, where there is seagrass habitat sensitivity and local anthropogenic pressure data, the impact of stressors can be integrated, further refining outputs (*Hu et al., 2021*). To add value to the model and support actionable results, the data needs to be robust, up-to-date and accurate. Finding accurate and reliable marine environmental data can be difficult (*Shepherd, 2018*), especially for coastal areas with complex hydrodynamics. Where there are data gaps, the collection of new data to generate new models may be required. Fine-scale hydrodynamic data and models have

been identified as especially important tools that could advance future seagrass habitat enhancement efforts (*Bertelli et al., 2022*).

Another future consideration for future seagrass habitat suitability modelling, is the use of appropriate habitat preference data. The habitat suitability mapping used in this assessment focussed on adult seagrass. However, if the seagrass habitat enhancement method being used relies on the germination of seagrass seedlings in areas where there are no existing adult seagrass shoots, different conditions conducive to seedling germination may be required. It is therefore important that future habitat suitability mapping considers the difference between environmental conditions that support seagrass seedling germination and conditions that support established mature seagrass plants (*Bertelli et al., 2022*). Research is still ongoing to determine the optimal conditions required for successful seed germination and establishment (*Infantes, Eriander & Moksnes, 2016*; *Unsworth et al., 2019*; *Cronau et al., 2023*). Once this evidence gap is filled, seagrass seedling habitat preference data could be used in habitat suitability models to provide outputs that are tailored to habitat enhancement methods that rely on successful germination.

## Complementary management and governance required to support investment into seagrass

The seagrass asset and risk register has highlighted that, for subtidal seagrass habitat assets within Plymouth Sound, condition is unfavourable, and the confidence in delivery of ecosystem service benefits remains insufficient. It also describes the level of confidence of the result, indicating where there is a need for more scoping and monitoring prior to the development of the marketplace for ecosystem services derived from seagrass habitat enhancement. Based on the range of data that contributes to the risk register, high risk ratings appear to be primarily driven by anthropogenic activities that result in physical abrasion of the seabed and poor water quality. It is essential that where appropriate, additional complementary governance and management measures that address dominant pressures are identified and applied to improve the confidence in delivery of ecosystem service benefits. Without the necessary measures in place, a seagrass habitat enhancement project faces increased likelihood of failure, from an ecological and socio-economic perspective.

Under current governance, the majority of the seagrass (78%) in Plymouth Sound is within a Marine Protected Area (MPA), though this nature conservation designation does not guarantee that the seagrass bed is protected from damaging activity. This is not uncommon and the ineffectiveness of MPAs to adequately protect seagrass habitats has previously been reported (*Eriander et al., 2017*; *Strachan, Lilley & Hennige, 2022*). All subtidal seagrass assets in the area are protected from bottom towed gear, which is particularly damaging to seagrass (*Erftemeijer & Robin Lewis, 2006*), under fisheries byelaws (*Cornwall IFCA, 2023*; *Devon and Severn IFCA, 2023*). That said, all seagrass assets in Plymouth Sound are not protected from other methods of potentially damaging fishing such as potting (*Henly, 2021*; *Cornwall IFCA, 2023*; *Devon and Severn IFCA, 2023*). Although potting is considered low impact compared to bottom towed gear, potting still

has the potential to damage seagrass habitat (*Marbà et al., 2004*; *Henly, 2021*; *d'Avack et al., 2022a*).

Confidence in ecosystem service delivery could be increased by enabling the effective protection of seagrass beds that goes beyond international directives and national and local regulations. Mechanisms by which this is achieved could include voluntary no-anchor zones and anchoring alternatives such as Advanced Mooring Systems (AMS) which have been shown to limit physical abrasion of the seabed (*Luff et al., 2019*; *Parry-Wilson et al., 2019*; *Marine Management Organisation (MMO), 2023*). Each protection mechanism has its own advantages and disadvantages, and these should be critically evaluated in the context of the local site before determining a single or combined approach. For the study site, a voluntary no-anchor zone (VNAZ) and the use of AMS systems has been achieved at small scales (*Marine Conservation Society, 2022*; *ReMEDIES, 2022b*). However, according to the latest statutory assessments, subtidal seagrass across the site appears to remain in unfavourable condition (*Natural England, 2023*). For AMS to become a viable solution for seagrass habitat enhancement it likely needs to be the exclusive or near exclusive means of mooring in a seagrass area, as the gains achieved using individual AMS can be negated by traditional moorings that abrade the seabed. Alternatively, VNAZs are required to restrict anchoring activity. With scaled up and effective protection there is potential to reverse the trend towards unfavourable status, improve the extent and condition and increase the confidence in the delivery of ecosystem service benefits for (prospective) buyers.

Evidently, other governance and management actions across important areas, such as water quality, will also be required to reduce the pressures acting on seagrass habitat assets. Although the relationship between water quality and seagrass is complex, it is generally understood that poor water quality and clarity impact seagrass health (*Jones, Hiscock & Connor, 1999*; *Burkholder, Tomasko & Touchette, 2007*; *Howard-Williams, 2022*). Poor water quality has been identified as one of the main causes of large scale seagrass loss in Europe over the last century (*de los Santos et al., 2019*). In the Plymouth Sound area, water body status data, and combined sewage overflow (CSO) data, show that water quality targets are not being met (*Environment Agency, 2021*; *DEFRA, 2022*). However, it is challenging to determine how much of an impact this has on the condition of local seagrass habitats and their ecosystem services. Research in Europe has shown a reversal of decline and the recovery of seagrass following major improvements in water quality (*Riemann et al., 2016*; *de los Santos et al., 2019*; *Krause-Jensen et al., 2021*). Improvements in overall water quality, particularly in water clarity and nutrient levels, driven by a holistic approach that considers upstream and downstream pollution within the catchment would be beneficial.

Fundamentally, complementary governance and management that mitigates harmful pressures to seagrass habitat are important for supporting successful investment into seagrass habitat enhancement. This is especially important for any financial product that uses a verified and quantified metric such as carbon sequestration or biodiversity uplift. Without adequate protection, the real carbon or biodiversity benefit derived from financed seagrass enhancement could be lost, even after the sale of a seagrass product or credit.

Effective protection will also be critical for any financial product that is intending to generate seagrass products or credits *via* the abatement of pressures.

## Exploring revenue models for seagrass habitat enhancement

Payments for Ecosystem Services (PES) refers to the payment for specific outcomes and benefits of nature and specific ecosystems. This approach has gained increasing attention as a means for paying for the restoration of a wider variety of habitats where specific economic benefits (such as carbon sequestration *via* offsets) have been identified, quantified, and verified. The key benefit of this approach is the perceived ability to unlock new funding streams (and even, in some cases, financing) to unlock delivery of more restoration than can be delivered by grants alone. This could positively contribute to the finance gap for nature, which has been estimated to be £56 billion for the UK between 2022 and 2032 (*Rayment Consulting, Green Finance Institute, Eftec, 2021*). This approach aims to recognise the services performed by nature, contributing $125 trillion to the global economy each year (*Costanza et al., 2014*). Ecosystem service markets offer a new channel for funding projects; however, this funding typically is paired with grants in an approach known as blended funding in which both grant and revenue streams can be used.

This approach, which so far has only been developed for seagrass projects in a couple of instances (*Oreska et al., 2020*; *PLAN VIVO, 2023*) has the potential to support seagrass habitat enhancement projects where there is a lack of sufficient governance and management to protect existing habitat (*Shilland et al., 2021*). There are a number of ecosystem service markets, many of which link closely to those services delivered by Seagrass habitats in the UK. The most developed ecosystem service markets are those for Carbon and Biodiversity offsets, but an emerging framework of payments for other ecosystem services exists, including the option to sell a group or bundle of ecosystem service benefits together. The creation of products linked to ecosystem service supply require a high degree of integrity including core principles to demonstrate the benefits of (*The Nature Conservancy, 2022*):

- Additionality: project delivers benefits beyond what would have been achieved without the incentive created from ES sales.
- Verifiability: ensure projects measure baselines and impacts accurately using robust methodologies and where applicable meet existing standards and protocols.
- Durability: ensure the durability of benefits is maximised and appropriate steps are taken to mitigate against risks that benefits are reversed.
- Transparency: ensure transparency through robust governance arrangements and a commitment towards public disclosure to maintain accountability.
- Do no harm and seek co-benefits: ensure projects adopt a holistic and integrated approach and seek co-benefits for nature and communities wherever possible.

The ability of an ecosystem service project/product developer to offer most or all of these benefits is critical to the credibility of the project, and the saleability of the ecosystem services generated. As demonstrated in this research, much of the information to understand the asset, the ecosystem services and the risk under current governance is driven by

environmental and societal demands, rather than any economic benefit that may be derived from the protection, restoration or creation of an ecosystem. From an ES product development perspective where projects are unable to show these benefits, they may be less attractive to a prospective buyer. For example, an unverifiable carbon credit offers limited certainty to a buyer seeking to reputably offset their carbon impacts. By using a natural capital asset and risk register approach, one can improve the understanding of how to offer these core principles, in particular confidence in the longevity and durability of the project, bringing closer the development of suitable ecosystem services markets for seagrass habitat enhancement projects. Overall governance of the assets may need to pivot to considering these ecosystems not simply as nature, but as natural capital and establish more effective monitoring to ensure no net loss of value.

## CONCLUSION

To support the development of sustainable investment options for Plymouth Sound seagrass conservation, we have developed a Seagrass Natural Capital Asset and Risk Register. This iteration of the framework is designed primarily for groups involved with seagrass habitat enhancement opportunities within Plymouth Sound. The method is highly transferrable and can be applied to support a natural capital approach to management in any marine region where similar data is available. As demonstrated by this study, this includes marine habitat enhancement projects that are looking to become self-sustaining and participate in ecosystem service markets. The asset register described in this report identifies seagrass habitat assets within Plymouth Sound and determines their current extent, condition, and spatial configuration. The risk register relates the status and trends of assets to the intended policy targets to assess whether there is a 'risk' of losing expected ecosystem services. The asset and risk register demonstrates that there is low confidence in the delivery of ecosystem services from seagrass beds in the Plymouth Sound area and that there is a risk of loss. Evidence from the risk register also usefully indicates the likely factors driving this increased risk of ecosystem service loss. In Plymouth Sound it appears that the most dominant adverse pressure to subtidal seagrass habitat are physical abrasion of the seabed and poor water quality. In order to ensure the development of ecosystem service markets for sustainable seagrass habitat enhancement there is a requirement for: more effective protection (beyond national and local regulations) and mitigation of dominant pressures; robust and frequent monitoring; and the use of financial products that integrate and develop credibility for this new means of funding.

## ACKNOWLEDGEMENTS

I would like to thank all the co-authors for their input into this work, in particular, Matthew Ashley, Tom Mullier and Sian Rees from the University of Plymouth Marine Natural Capital team. The authors would like to thank Ollie Thomas from the University of Plymouth and Mark Parry from the Ocean Conservation Trust for providing spatial data that contributed to the composite seagrass habitat layer. Finally, the authors would

like to thank the PeerJ topic editor and reviewers for their constructive comments in the review process of this manuscript.

### Funding

This work was funded by the Natural Environment Investment Readiness Fund (NEIRF). The funders had no role in study design, data collection and analysis, decision to publish, or preparation of the manuscript.

### Grant Disclosures

The following grant information was disclosed by the authors:
the Natural Environment Investment Readiness Fund (NEIRF).

### Competing Interests

Amelia Sturgeon is the coordinator for the Tamar Estuaries Consultative Forum. Zoe Sydenham is an employee at Plymouth City Council. Mark Parry is an employee at the Ocean Conservation Trust. Katey Valentine is an employee at BeZero Carbon.

### Author Contributions

- Guy Hooper conceived and designed the experiments, performed the experiments, prepared figures and/or tables, authored or reviewed drafts of the article, and approved the final draft.
- Matthew Ashley conceived and designed the experiments, performed the experiments, prepared figures and/or tables, authored or reviewed drafts of the article, and approved the final draft.
- Tom Mullier conceived and designed the experiments, performed the experiments, prepared figures and/or tables, authored or reviewed drafts of the article, and approved the final draft.
- Martin Attrill analyzed the data, authored or reviewed drafts of the article, and approved the final draft.
- Amelia Sturgeon analyzed the data, authored or reviewed drafts of the article, and approved the final draft.
- Zoe Sydenham analyzed the data, authored or reviewed drafts of the article, and approved the final draft.
- Mark Parry analyzed the data, authored or reviewed drafts of the article, and approved the final draft.
- Katey Valentine analyzed the data, authored or reviewed drafts of the article, and approved the final draft.
- Sian Rees conceived and designed the experiments, performed the experiments, authored or reviewed drafts of the article, and approved the final draft.

### Data Availability

The raw data is available in the Supplemental File.

## Supplemental Information

Supplemental information for this article can be found online at http://dx.doi.org/10.7717/peerj.17969#supplemental-information.

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
