# Peer review of "Using a natural capital risk register to support the funding of seagrass habitat enhancement in Plymouth Sound"

_PeerJ, doi:10.7717/peerj.17969_

## Round 0.1 · original submission · Minor Revisions

Thanks for your submission. I agree with the reviewers request for minor revisions. Please elaborate on the following:
- introduction (threats to seagrass and impacts to benefits they provide)
- habitat suitability modeling (limitations, assumptions, and omitted variables)
- scientific and policy relevance to the global seagrass conservation community
- transferability of this approach to locations outside of the UK

Reviewer 1 ·

Basic reporting

The writing style is very solid, clear and unambiguous throughout and the manuscript is well-polished and put together to a high standard. Figures and tables are informative and well-constructed and the context provided and literature cited was generally excellent.

I did have to re-read to introduction as it feels like the justification for using the asset and risk register could be more clearly stated there. At present, the rationale of the introduction reads … valuing ES is a potential source of funding to support seagrass conservation, the natural capital approach incorporates risk assessment and this is used to assess estuarine habitat persistence. There is a paragraph on NCA (assets) but not how risk could be quantified and inform/guide revenue generation. Demonstrating the types of risks seagrass habitats face and the impact they could have on ES outside of the first paragraph would close the loop here.

Experimental design

I am confident that the risk assessments undertaken by the authors are robust and well-considered based on available data. However, I have a few issues with the habitat suitability mapping. The assumptions around certain parameters used to predict suitability (eg. Exposure and depth) need defining to parameterise the model. Some environmental variables like turbidity and Nitrogen were not considered but form part of the risk assessment, which means it sits almost distinct from the main part of the study. If the idea is to ignore the variables that are potential threats when carrying out the analysis this should be stated clearly and perhaps the limitations of the analysis clearly defined when presented in the methods section. Light availability and hydrodynamics are mentioned as potential features of habitat modelling in the discussion and again these can be included in a robust habitat suitability model but probably need to feature in the methods when the approach is introduced. Finally, it’s a little odd that areas adjacent to existing seagrass were not considered given the ReMEDIES restoration project carried out in the SAC at Jennycliff only considered expansion of existing seagrass beds as this offered the best chance of success. Overall, this part of the study felt less confident than the rest.

Validity of the findings

The outcomes of the risk analyses were telling and were explained well in the results and discussion. It is a shame that the risk assessment for almost half of the total seagrass coverage in the SAC (intertidal) has such low confidence. I wonder if this could be addressed with further research on case studies? It would add weight to the findings certainly if the reader was provided more information about the risks facing intertidal seagrass beds in section 2.6.2 asides the algal cover mentioned and possibly connections explored with nutrient addition as demonstrated as a threat for intertidal Z. noltei in the Wadden Sea by Dolch et al 2013 (https://doi.org/10.1016/j.seares.2012.04.007). If this is beyond the scope of this study, the lack of available data on condition assessment by Natural England could be highlighted in the methods section mentioned above.

As mentioned above, the value/limitations of the habitat suitability modelling based on the points above should be mentioned for completeness in the results.

Figures 4 and 5 in S1 are not displaying in the review copy so I could look at them.

Additional comments

The message of the paper is an important one: it is irresponsible to consider the monetary value of ES for seagrass without high confidence estimation of the risk that those ES are not about to be lost. The paper clearly demonstrates there is a good chance they are indeed in decline in the Plymouth Sound as the risk analysis outcomes are compelling for subtidal seagrass. Valuable suggestions as to how these could be measured are then offered in the discussion. Realistic identification of the issues and framework for monetarization followed that were also informative.

Finer text errors I picked up:
Line 92 extra ‘a’ or missing ‘meadow’
Line 341 extra full stop
Table 2 not referred to in text
Line 478 extra ‘to’
Line 585 extra.

Reviewer 2 ·

Basic reporting

no comment

Experimental design

Did you select those policies based on its relevance to seagrasses? to UK? Please elaborate this in section 2.8

Validity of the findings

Who are the target stakeholders for this Seagrass Natural Capital Asset and Risk Register? How can this be applied in other areas with similar seagrass conditions?

---

## Round 0.2 · accepted · Accept

Thank you for addressing the review comments. This manuscript is ready for publication. Thank you!

Reviewer 2 ·

Basic reporting

no comment

Experimental design

no comment

Validity of the findings

no comment